# The Design and Application of Game Rewards in Youth Addiction Care

**Marierose M. M. van Dooren** [1,*]**, Valentijn T. Visch** [1] **and Renske Spijkerman** [2]

[1]   Faculty of Industrial Design Engineering, Delft University of Technology, 2628 CE Delft, The Netherlands;
      v.t.visch@tudelft.nl

[2]   Parnassia Addiction Research Centre (PARC), Brijder Jeugd, Parnassia Group, 2512 HN The Hague,
      The Netherlands; renske.spijkerman@brijder.nl

*   Correspondence: m.m.m.vandooren@tudelft.nl

**Abstract:** Different types of rewards are applied in persuasive games to encourage play persistence of its users and facilitate the achievement of desired real-world goals, such as behavioral change. Persuasive games have successfully been applied in mental healthcare and may hold potential for different types of patients. However, we question to what extent game-based rewards are suitable in a persuasive game design for a substance dependence therapy context, as people with substance-related disorders show decreased sensitivity to natural rewards, which may result in different responses to commonly applied game rewards compared to people without substance use disorders. In a within-subject experiment with 20 substance dependent and 25 non-dependent participants, we examined whether play persistence and reward evaluation differed between the two groups. Results showed that in contrast to our expectations, substance dependent participants were more motivated by the types of rewards compared to non-substance dependent participants. Participants evaluated monetary rewards more positively than playing for virtual points or social rewards. We conclude this paper with design implications of game-based rewards in persuasive games for mental healthcare.

**Keywords:** gamification; play persistence; reward types; addiction; youth; persuasive game design

## 1. Introduction

An evidence-based therapeutic strategy to motivate substance dependent individuals to remain abstinent is to add monetary-based rewards to evidence of successful behavioral change, e.g., substance-free urine tests [1]. Rewards can be seen as strong motivators to influence behavior change [2–5] and are a crucial aspect in the design of entertainment videogames to enhance not only feelings of enjoyment and flow [6] but also feelings of mastery, autonomy, and a sense of belonging [7]. Because it takes time for users to intend, start, and maintain behavior change, game elements, such as rewards, have often been used as motivational ingredients in Persuasive Game Design (PGD) [3,8,9].

The aim of PGD is to transport the users' real-world experience towards a (partial) game world experience that is more enjoyable and engaging than real-world experiences [10–12], thereby enhancing persistence of specific desired behavior in the real world, known as the transfer-effect [13,14]. Research has shown that applying game elements in a serious context can positively influence health-related problems and behaviors [15,16], such as anxiety management [17], physical therapeutic exercise and fitness [18,19], burn pain management [20], diabetes [21], and asthma [22]. However, research has also suggested that applying game elements in a serious context can reduce overall engagement and intrinsic motivation [23] or lead to unintended effects that distract players and lowers the overall effectiveness of an intervention [24]. Most importantly, game elements use extrinsic rewards, such as levels and points, to enhance engagement of users, while striving to enhance users' general feelings

of competence, autonomy, and a sense of belonging and connectedness with others [10]. These three elements form the basic human psychological needs that facilitate users' motivation, both intrinsic and extrinsic, to execute specific behavior [7].

Thus, PGD seems to be fruitful for enhancing positive healthcare effects, since it helps players to aim for a given target experience or behavior. Crucial in the persuasive effect of a game design is the choice of the used game elements. These are the elements within a game that function as core motivators for a play experience, such as *a challenge* in platform games, *social teaming* in soccer sport games, or *exploration* in role-playing games. Among these game elements, *rewards* are one of the most applied kind of elements. Sometimes rewards are designed as a core game-element in a game, such as the monetary rewards in gambling, and sometimes as a supportive game element, such as the weapons and powers you can earn as a reward for completing a challenge in MMORPGs. Although motivational effects of rewards in daily life have been studied extensively in psychological and neurocognitive studies [14,25–29], there is surprisingly little fundamental research about the motivational effects of rewards in games.

In games, rewards are most typically applied in the form of monetary rewards, virtual points, and social rewards [5,30]. These three reward types differ in their value of use. Monetary rewards have a dominant value in the real world outside the game. In contrast, virtual points have their dominant value within the game world, and social rewards, such as received compliments about your gameplay by your playmates, have a value in both the real world and the game world [31]. Monetary rewards consist of a tangible amount of money that a player receives for a specific performance [32,33]. Virtual points are used as a scoring system or as a way to buy virtual goods that are usable in the game (e.g., better weapons). Scoring systems based on the earned player points are often a symbolic way of reflecting the players' progression, performance, achievement, and competence [32]. In social rewards, players give and receive compliments to and from other players, or they invite and are invited to join specific player groups. This type of reward includes positive incentives related to the general human need of feeling related to others [32] and receiving social recognition for specific behaviors [34,35]. From a neurocognitive perspective, preliminary findings from Functional Magnetic Resonance Imaging (fMRI) research suggest that these three reward-types may activate specific areas in the brain [36]. For example, brain areas that have been linked to the processing of self-related and social information showed more activation when social rewards were gained than monetary rewards or performance feedback, such as points.

In the present paper we will investigate, for the purpose of serious game design, the motivational difference of the three basic types of game rewards: monetary rewards, social rewards, and virtual points. The application of game elements (such as game rewards) in a non-entertainment (i.e., "serious") context is called "gamification" [37]. In order to study the application value of game rewards in serious contexts comprising specific user groups, we involved (a) adolescent patients with substance use disorders from a substance addiction care context and (b) a same-aged control group of high-school students without substance use disorders. The context of substance addiction therapy might benefit from the study of persuasive game design involving rewards, since reward-based game behavior and substance-use both derive their motivation from shared neurological dopamine systems. More specifically, video gaming is associated with dopamine release, and all addictive substances trigger increases in dopamine in a key region of the reward (limbic) system in the brain [38,39]. Additionally, adding game-elements to an addiction therapy might make the therapy more engaging for patients, and hence enhance the therapeutic adherence [40]. While game-rewards may be particularly motivating for adolescents, it is not clear whether this also holds for adolescents with substance use disorders. Neurological findings suggest that—compared to non-dependent persons—the application of rewards may have less impact on substance dependent individuals due to a hyperactive dopamine system for psychoactive substances (alcohol, amphetamine, opiates, or marijuana) and a decreased sensitivity to stimuli that are not related to these substances [39,41–46]. This "dampened" effect of non-substance related rewards in substance dependent persons informs our hypothesis that game-rewards may have a lower motivational effect in

this population than in a non-dependent high-school population. Although we do have evidence that rewards can work in the clinical practice of addiction treatment—particularly when using monetary incentives following an evidence-based contingency management scheme [47–50]—neurocognitive findings indicate that natural rewards may have a lower impact on this population. It is unclear whether substance dependent individuals will be sufficiently motivated by game rewards, since this type of individual may be more strongly motivated by the expected rewarding effect of substance use. To determine which types of rewards are suitable for persuasive games aimed at patients in mental health care, it is important to empirically test the potential impact of game rewards for specific patient groups, such as individuals with substance use disorders. The present study will, thus, focus on comparing the effects of the three basic separated reward types between a clinical sample of substance dependent adolescents and a control group of non-dependent high-school students. Because substance dependent individuals may show decreased sensitivity to rewards [39,41–46].

We hypothesize that all separate reward types will be less motivating for them compared to their non-substance dependent counter-parts.

## 2. Materials and Methods

### 2.1. Ethics

The Medical Ethical Committee of the Leiden University Medical Centre in the Netherlands granted exemption for a full ethical application.

### 2.2. Participants

Participants (aged between 12–24 years) were recruited from two locations in the Netherlands. A total of 32 non-substance dependent adolescents were recruited from a secondary school and 36 substance dependent adolescents were recruited from a substance addiction care facility. Due to computer problems during the test, we had to exclude 23 participants (16 substance dependent and 7 non-dependent adolescents). Approximately 50% of these participants ($N = 11$) did not play the game for all three types of rewards because of software problems. The other 50% of these participants ($N = 12$) unwillingly pressed the stop-button while playing, even though they did not want to stop playing the game. At the start of the experiment we clearly explained to participants that they could press the stop-button if they wanted to stop playing the game (see Figure 1). This was important for our analysis, since the stop-button was directly related with the dependent variable "play persistence". However, since participants pressed the stop-button even though they did not want to stop playing, either they did not understand this explanation or they pressed the button by accident. When participants did not play the game for all three types of rewards, we had to exclude them from the whole study as we could not compare their play persistence for the different types of rewards anymore.

The final study sample consisted of 45 participants, with 20 substance dependent and 25 non-dependent adolescents. The group of substance dependent adolescents contained fewer females (15%) compared to non-dependent adolescents (52%), matching the general substance dependence population that also consists of more males [51,52]. We did not collect personal information regarding the type of substance dependence, since this was not the focus of the study. In addition, it was often comorbid and asking for this information might have decreased the participants' motivation to engage in the playtest study. Adolescents in Dutch addiction care most often receive therapy for cannabis, alcohol, and gaming. A smaller group receives therapy for simulants (mainly amphetamine, but also cocaine or ecstasy) [53–55]. We tried to match the age of both substance dependent and non-dependent groups. The average age of the respondents from the secondary school was around 16 years old (14–18 years old), and patients who are in therapy at the youth addiction care clinic are generally around 18 years old (12–22 years old) [54].

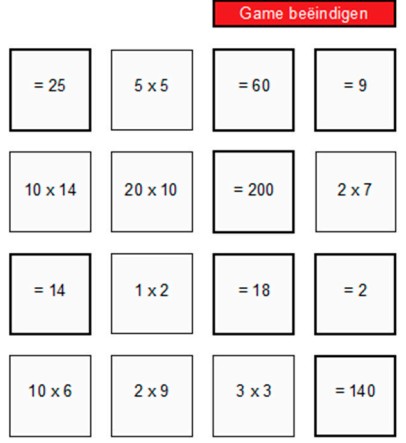

**Figure 1.** The tablet-based game showing the game-task to combine multiplications and outcomes. Translation of text in red button upper right: "Ending the Game". Text below states: "Get another 5 correct combinations to receive 5 more points! You already have: 25 points".

*2.3. Design*

Per type of reward, participants were able to spend a maximum of 40 min playing the game. If participants used the maximum playing time with all types of rewards, the maximum playing time would be two hours. Participants in current study played the game with all types of rewards in a total of 30–60 min. The game consisted of a four by four grid with 16 buttons. Of these buttons, 8 randomly displayed multiplications of 2 up to 9, and the other 8 displayed possible answers of the multiplication products. Of these 8 possible answers of the products, 6 matched the outcomes and 2 were incorrect. Participants were instructed to match a multiplication product, and after each match the screen refreshed.

Participants received incremental rewards after a specific number of correct answers (after 3, 6, 9, 12, 15, 18, 21, 24, 27, 30, 33, 36, 39, and 42 correct answers). The screen showed how many correct matches were needed to obtain the next reward and how many rewards the participant had already earned. Participants could, thus, earn a total of 14 rewards per session and complete a maximum number of 315 products if they played the maximum play time and always answered correctly. During the whole game, the screen showed a "stop-playing" button at the top of the screen. This provided the participant with the possibility to stop playing the game at any moment when preferred (see Figure 1). After hitting the "stop-playing" button, a new game started with similar exercises but with another randomly chosen different reward type until the player had played for all three reward types. At the end of the study, all participants received 10 euros for their participation, regardless of their score in the game. The participants were not informed about the participation fee beforehand.

In total, participants played three game-sessions. In each session they would play for one of the three reward types: monetary rewards, virtual points, or a social reward (see Figure 2). Regarding the monetary reward, participants could receive 50 Eurocent per reward until they reached a total of 7 Euros. They received this reward type after the study. Regarding the virtual points, participants could receive 5 points per reward until they reached a maximum of 70 points. The third reward consisted of a social reward, where participants saw a pop-up picture of a randomly selected blurry face, with a thumbs up and a textual compliment. The blurry faces were taken from a pool of portraits of participants of the study that we photographed before starting the study. For ethical considerations we blurred the photographs to the extent that faces known to the participant were recognized but faces unknown to the participant were not. Participants received one compliment per reward moment, which could vary according to five different kinds of texts: "Well done!", "Wonderful!", "How smart!", "Calculation tiger!", "Thumbs up!".

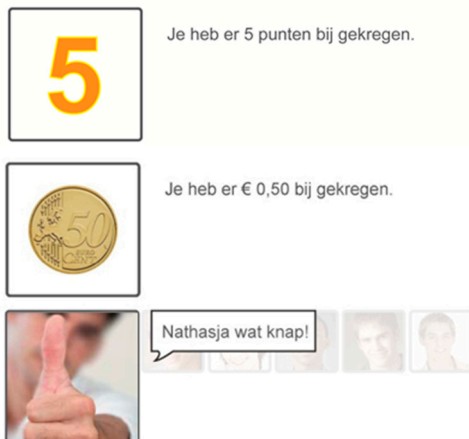

**Figure 2.** Examples of three types of rewards (translation from top to bottom: "You earned 5 more points", "You earned 50 more Eurocents", "Natasha, how smart!").

*2.4. Variables and Measures*

As the independent variable we used the type of reward (monetary rewards, virtual points, and social rewards) and reward evaluation was considered as the dependent variable. We used time in minutes that users spent playing the game, until they hit the "stop-playing" button, as a measure of play persistence (time spent playing as a measure of persistence was also used in a previous study [56]). Participants could evaluate the reward by answering the following four questions on a five-point Likert scale (0 (= totally disagree), until (4 = totally agree)): (1) "I did not want to quit while winning/earning "the reward type" (only fill in if you have stopped before the end of the test)"; (2) "I wanted to continue playing because of "the reward type"; (3) "I think that "the reward type" is a good reward"; (4) "I am happy with the amount of "the reward type" I have won".

*2.5. Procedure*

Participants first provided written informed consent for study participation, after which they received an iPad for use in the present study. At the start of the game, participants filled in their name and were instructed as a practice to first complete as many multiplications as possible within two minutes. After that, participants received information about how the game worked. They were also given the opportunity to ask questions if anything was unclear. If there were no questions or all questions were answered, the game started. After the third and last game-session, participants were asked some final questions about playing the game in general. For each respondent, all game sessions took place during one session, in which the order of the types of rewards was randomized.

## 3. Results

*3.1. Strategy of Analysis*

All analyses were conducted in SPSS version 22. Since the data were not normally distributed, as shown by a Kolmogorov-Smirnov test, we applied nonparametric tests. Without the first item, Cronbach's Alphas for the evaluation of monetary rewards, virtual points, and social rewards were respectively 0.84, 0.85, and 0.82.

*3.2. Manipulation Checks*

According to the nonparametric independent samples tests there was no statistically significant difference between our control variable "order of reward type" and time participants played with the rewards (all *p*-values > = 0.20). Furthermore, there was no significant difference between the "order of reward type" and reward evaluation of all types of rewards (all *p*-values > = 0.29).

### 3.3. Difference in Playing Time According to Reward Types between Substance Dependent and Non-Dependent Adolescents

To test differences in playing time we conducted a GEE-analysis (Generalized Estimating Equations), including playing time as the dependent variable, reward type as the within-subject variable, group (substance dependent vs. non-dependent) as a factor, and gender as a covariate (see Table 1). The significant effect of reward type (monetary, social, and virtual points) indicated that adolescents played longer for monetary rewards ($M = 24.35$, $SD = 11.39$) compared to social rewards ($M = 9.30$, $SD = 10.22$) or virtual points ($M = 12.06$, $SD = 11.15$). Results further showed significant effects for the factor group (substance dependent vs. non-dependent) ($X^2 = 13.77$, $p < 0.05$) and the covariate gender ($X^2 = 11.84$, $p < 0.05$). Regardless of type of reward and gender, adolescents with substance dependence ($M = 18.14$, $SD = 14.23$) played longer compared to non-dependent adolescents ($M = 12.91$, $SD = 10.84$). The significant effect of gender suggested that girls ($M = 16.83$, $SD = 12.27$) played longer compared to boys ($M = 14.36$, $SD = 12.89$), regardless of group and type of reward.

**Table 1.** Regression estimates for play consistency.

|  | B | SE | Wald $X^2$ (95% CI) | Sig |
|---|---|---|---|---|
| Virtual points | −12.29 | 1.91 | 41.37 (−16.04 to −8.55) | 0.000 |
| Social rewards | −15.05 | 2.26 | 44.40 (−19.47 to −10.62) | 0.000 |
| Monetary rewards | . | . | . | . |
| Substance dependent | 7.20 | 1.94 | 13.77 (3.40 to 11.01) | 0.000 |
| Non-substance dependent | . | . | . | . |
| Gender | 5.35 | 1.55 | 11.84 (2.30 to 8.40) | 0.000 |

### 3.4. Difference in Reward Evaluation According to Reward Types between Substance Dependent and Non-Dependent Adolescents

In a second GEE-analysis with reward evaluation as a dependent variable, we tested the effects of reward type and group while controlling for the covariate gender. The type of reward was the only significant variable ($X^2 = 30.61$, $p < 0.05$). Adolescents evaluated playing for monetary rewards ($M = 3.02$, $SD = 1.00$) significantly more positively than playing for virtual points ($M = 2.22$, $SD = 1.03$) or social rewards ($M = 2.35$, $SD = 1.03$) (see Table 2).

**Table 2.** Regression estimates for reward evaluation.

|  | B | SE | Wald $X^2$ (95% CI) | Sig |
|---|---|---|---|---|
| Virtual points | −0.88 | 0.17 | 28.38 (−1.21 to −0.56) | 0.000 |
| Social rewards | −0.66 | 0.16 | 16.64 (−0.97 to −0.34) | 0.000 |
| Monetary rewards | . | . | . | . |
| Substance dependent | −0.02 | 0.19 | 0.01 (−0.39 to 0.36) | 0.93 |
| Non-substance dependent | . | . | . | . |
| Gender | −0.04 | 0.18 | 0.05 (−0.40 to 0.31) | 0.82 |

### 3.5. General Results

Results showed that there was a statistically significant difference in playing time according to reward types. Participants played significantly longer when they were playing for monetary rewards compared to the other types of rewards. In addition, there was a statistically significant difference in participants' reward evaluations of the game according to reward type, and participants evaluated playing for money more positively compared to the other types of rewards. When comparing substance dependent and non-dependent participants, results showed that substance dependent participants played longer compared with non-dependent participants. In addition, regardless of type of reward, female participants played longer compared to male participants.

## 4. Discussion

In the present study we tested if the effects of three types of rewards (social, monetary, and virtual) on game play duration, and game evaluation differed between substance dependent versus non-dependent adolescents. Adolescence is a period in life that is characterized by increased risk taking, resulting from an overactive reward system in the brain [57], relative to childhood and adulthood [58,59]. Therefore, rewards may have an increased motivating effect on adolescents and can be used as a useful incentive. However, it was unclear if, and which, game-based rewards would work in a substance addiction therapy context, based on the link found between a hyperactive dopamine system and a decreased sensitivity to natural rewards in substance dependent individuals [39,41–46].

Our findings suggest that rewards can successfully motivate both substance dependent and non-dependent adolescents to continue their interaction with a game. When users interact more or for longer with a game, it is more likely that the transfer effect of the game will be achieved. Therefore, our findings confirm that rewards may successfully be applied in persuasive game design for both substance dependent and non-dependent adolescents to enhance motivation for tasks (e.g., therapy adherence). However, this study only focused on the effects of rewards on serious tasks and not therapeutic tasks. With serious tasks there is a direct interaction between rewards and behavior, but with therapeutic tasks the point of impact generally takes more time. In addition, in persuasive game design for therapeutic tasks it is needed to carefully match the rewards with the desired transfer effect in order to avoid confounding conflicts between the two, and to also study contributions to long-term therapy effects. This study shows that with serious tasks, rewards are suitable to enhance motivation to continue interaction with a product. More research is, however, needed to see if rewards are also effective for therapeutic tasks with a more long-term effect.

Our results further indicate that when receiving rewards, substance dependent adolescents played significantly longer than non-dependent adolescents. Both groups of adolescents did not differ in how they evaluated the reward types. Overall, adolescents evaluated monetary rewards more positively compared to the other types of rewards. An explanation for this might be that game play duration was evoked by other (perhaps unconscious) processes or triggers that were not strongly linked to the explicit evaluation of all three types of game rewards. For example, substance dependent adolescents may experience their clinical "real world" context as less exciting and playful than how non-dependent adolescents experience their non-clinical real world. In terms of the persuasive game design model [12], the starting position of the participants with substance dependence might, thus, be positioned more towards the real world than the starting position of the high-school participants. This difference might influence the motivational effect of the designed mathematical game in transporting the user's experience towards a game world. The motivational effect of a game might be stronger in a less playful environment than in an already playful environment. Future research has to be conducted to investigate this relationship between experienced real world position, effect of game, and its resulting game world experience.

The finding that participants with substance dependence played longer for the types of rewards was contrary to our expectations. We expected that participants with substance dependence would play shorter for any reward type during the experiment, as research showed that substance dependent individuals have an overall decreased reward sensitivity [39]. This previous hypothesis was confirmed in previous research by Kim et al. (2014), who compared the motivational effects of similar reward types, i.e., performance feedback, social rewards, and monetary rewards, between internet addicted adolescents and non-addicted adolescents. The outcomes of this particular study did suggest a decreased sensitivity to game rewards in participants with an internet addiction compared to non-addicted participants [45]. Our finding on the impact of monetary game rewards are in line with previous research showing that monetary incentives have successfully been applied in substance abuse therapy [47–49]. For virtual points and social game rewards our findings cannot be confirmed by previous clinical research, although some forms of evidence-based therapies do apply to these types of incentives to reinforce non-drug related activities.

This study has some limitations that need to be mentioned. First, we did not differentiate the group of substance dependent adolescents according to their main type of substance problem, e.g., alcohol, cannabis, or stimulants, nor did we differentiate groups according to specific personality characteristics. Some studies have shown that different player groups, i.e., groups with different personality dimensions, can be more interested in, or motivated by, specific game-rewards than others [60]. Since studies have found that some personality traits are more associated with substance addiction than others, more research is needed to further explore this topic [61]. Secondly, although we knew the age range of patients that were admitted to the youth addiction care facility, we did not record the age of those who participated in our study and could not control for age as a covariate in our analyses. In addition, it is important to take into account the out-game value of rewards for users. Future studies should focus on the need for personalizing rewards and whether different player types, personality traits, and types of substances are linked to reward sensitivity [62,63]. Secondly, although we tried to keep the intensity of the three reward types comparable, i.e., either one compliment, 5 points, or 0.50 Eurocents per reward, it is not certain that we succeeded in this. It is possible that participants' reward experiences were affected by how the rewards were designed [64]. Future studies could address this issue by testing a more sophisticated differentiation in types and intensity of rewards.

## 5. Rewards in Persuasive Game Design: Implications

The present study investigated if game-based rewards can be used as motivating game-elements in a persuasive game for adolescents with a substance use disorder. The results turned out to be positive, since the types of rewards motivated substance dependent adolescents in addiction care more compared to non-dependent adolescents in high-school. Thus, a persuasive game designer developing eHealth for an addiction care context can consider using rewards to motivate patients. However, how rewards can best be applied in a persuasive game does not follow from our study. In the present section, we will provide suggestions for reward inclusion in persuasive games.

In persuasive game design practice, the choice for a motivating game element is not made at the start of a project. Following our Persuasive Game Design (PGD) method [40], gamification projects start by specifying the real world goal of a persuasive game, i.e., the "transfer effect", followed by investigating the "user context". The information gathered in these two stages is used in the next stage, the gamification design, which includes choosing and designing game elements for the game. The choice for the type, form, and interaction schedule of a reward will, thus, be influenced by the transfer goal and user context, as we will show in this section.

A transfer effect can be specified into four components (effect type, change type, point of impact, and domain), which all can influence the choice for a motivating game element. For instance, if the desired type of transfer effect in a persuasive game is to increase the social relatedness of employees on the work floor [65], a game designer might rather motivate the employees by social rewards, e.g., compliments, in the game instead of monetary rewards, which might lead to economic disparities among the employees and decrease social relatedness among them. In contrast, when the aim of a persuasive game is to increase self-efficacy among independent living elderly, monetary rewards might be considered as a central game element, since they can increase a person's required resources to overcome real-life obstacles, to make their own choices, and thus enhance confidence in personal capabilities [66]. Other types of transfer effects, like learning, might not favor rewards as central game elements but rather motivate users by providing challenges or exploration opportunities.

Next to transfer type, a transfer effect is specified by its *change type* (initiating, altering, diminishing, or reinforcing a behavior) and its *point of impact* (i.e., when one expects the transfer effect to occur—during gameplay (e.g., exergames), directly after gameplay (learning games), or a long time after gameplay (lifestyle change)) [40]. The expected point of impact of a transfer effect will influence design decisions regarding rewards. This will not so much influence what type of reward (social, monetary, points) will fit the persuasive game, but rather how a player can obtain a reward, i.e., the contingency of a reward design in a game. Rewards can be linked to the player's *tasks*, *performance*, or *engagement* [67]. For

short-term initiating transfer effects, such as physical exercise in an exergame, rewards can be linked to the task (get a reward when the player has completed 10 sit-ups), to the performance (a reward when the player does 10 sit-ups in a short time), or to engagement (a reward when the player has played the game for 10 min). Long-term effects, such as a lifestyle change, might favor engagement contingent rewards (a reward every week the player eats healthy and does physical exercise). One might also design combinations of reward contingency relations. For instance, in a persuasive game with a transfer effect to quit smoking, one might start to earn rewards by completing tasks, e.g., not smoking for one day, apply performance-contingency after a week, receive a reward when the player has not smoked and has been active in sports, and use engagement-contingency after a few months by earning a reward when the player still has not smoked.

Especially when a transfer effect has a medium- or long-term point of impact, it is crucial to avoid player acclimation [68] of a reward; players might attribute high value at a reward during the beginning of the gameplay but might not be motivated by the same reward later on in the game. To account for such a decrease of motivation by reward, a game designer can vary the process of giving the rewards. Variation in rewards to maintain player motivation can be achieved by (1) varying the contingency of the reward (see the quit smoking example above), (2) the value of the reward (for instance increase the value of a reward gradually or provide an reward with unknown value, such as a "mystery box") [68], or (3) inserting variable reinforcements [69], such as a sudden rewards occurring at unexpected moments during the gameplay.

The design decision for the form and placement of a reward in a persuasive game does depend on the specific transfer effect, but it will also depend on the user and context of use of a game. People can differ in their general response to rewards or they may be especially responsive to specific types of rewards. For example, compared to adults, adolescents appear particularly "reward-sensitive", and hence show stronger neural and behavioral responses to rewarding stimuli [64,70,71]. Other studies suggest that responsiveness to a specific type of reward may be linked to gender [72,73], personality traits, such as empathy or impulsivity [34], and mental disorder [74–76]. To optimize the design of PGD [77] and to develop the most suitable reward for a specific interaction of a specific individual, it is crucial to investigate the motivations and demographics of your target group. A useful method to tailor games to specific personality types is the Hexad framework [62]. This framework categorizes users into six types of player personalities loosely related to the Big Five personality traits: Disruptors (motivated by change), Socializers (motivated by relatedness), Philanthropists (motivated by purpose), Free Spirits (motivated by autonomy), Achievers (motivated by competence), or Players (motivated by extrinsic rewards). Although such a player type classification might work well to design entertainment games, the serious context of a persuasive game might crucially change the player type; someone might be a socializer in an entertainment game context but an achiever in a working context. Investigating if and how the playing motivation of a user differs in an entertainment and serious context is, thus, a crucial phase in the persuasive game design process and will influence design choices regarding rewards.

In the present study we used three basic types of game rewards (monetary, social, and point rewards). In game practice, and especially in entertainment games, other reward types are used as well, and they often occur in combinations. Schell (2008) lists a set of nine commonly used entertainment-based in-game rewards [68]. These include points and social rewards, such as praise, but also nested rewards that are provided when a player reaches a specific amount of points, such as prolonged play opportunity, unlocking a new level, perceiving a juicy spectacle, or improving character powers. Money, as a reward with an out-game value, also comes in variants, e.g., discounts or gift coupons. Just like the in-game rewards, these out-game rewards often are paced during the gameplay by points—a player has to collect in-game points and can only exchange a predefined amount of points into a reward with out-game value.

## 6. Rewards in Persuasive Game Design: Case Study

In a youth addiction care context, we involved patients and therapists in a Persuasive Game Design process aimed at realizing a transfer effect to enhance a patient's motivation to set and achieve cognitive behavioral therapy-related goals. To understand what game-experiences patients expected to be motivating, we used Playful Experiences (PLEX) cards representing 22 game experience categories [78]. The most motivating experience patients selected was the experience of "thrill" [79]. We then carried out brainstorm sessions with game designers from a serious game design agency in the Netherlands to generate the following game mechanics that we expected could motivate patients in a youth addiction care context [80]: risk taking, progression map system, selfie photograph feedback system, reward system, and personal values. These mechanics were evaluated by nine patients and eight therapists, who ranked them based on the expected motivational impact for the transfer effect. Interestingly, patients and therapists differed in their ranking. Patients rated risk taking and personal rewards as the best motivating mechanics, while therapists rated risk taking and external rewards as most favorable [80]. The preference of therapists regarding the external rewards seemed to correspond with current therapy techniques that already apply external rewards to patients by using contingency management [47]. However, it is essential for rewards to correspond with both the context of application, i.e., the addiction care context, and the preference of the end-user, i.e., the patient.

In order to optimize the motivational effect of a reward in a persuasive game, a game designer can tailor, as in the Personalized Design Process model [81], the reward as much as possible to the preference, type [82], or personality [83] of the end-user. Moreover, it is possible to design a game in which end-users can choose or generate their own rewards, or to let fellow players tailor the rewards for them. In our persuasive game design for a therapy context, patients did not find our pre-set reward (a 3D printed token of a goat that was related to the level they achieved) motivating. Therefore, we wanted to provide them with a reward for their accumulated points that would be more personally relevant and motivating. This resulted in giving patients the opportunity to choose their own reward in collaboration with their therapist. In addition, we aimed to increase the patients' therapeutic involvement in goal setting by using mechanisms similar to those used in the "shared decision-making" approach in therapy [84]. The rationale for this adaptation was based on patients' negative evaluations of the pre-set tasks in setting goals. According to the patients, this procedure made it more difficult for them to set goals which were sufficiently challenging, personally relevant, and valuable. In the adapted version, both the therapists and patients could decide on which long-term therapy-related goals they would use together. This ensured that these goals were relevant for the patient's health objectives and of intrinsic value to the patient. In addition, patients could type in their own short-term tasks. In sum, in our iteration we included three opportunities to personalize the game: reward, (main) goals, and short-term tasks. However, it can be debated how much personalization would be possible and preferable in game design. For example, would it be preferable to design one game for each individual user, or to design one game that is so open that it can be fully personalized to each individual user? In both situations one can ask if these games would have enough overlap to be considered as the same game resulting in the same comparable effect.

## 7. Conclusions and Future Research

Involving rewards as a basic game-element in persuasive game design to redesign psychotherapy has shown potential for youth addiction care, as substance dependent adolescents were more motivated by rewards compared to non-dependent adolescents. In the current study, participants received rewards based on a fixed reinforcement schedule. It would be interesting to explore different schedules for providing rewards, since specific users may prefer a variable schedule more than a fixed one, which can be used for personalization. In addition, it is interesting to study how the motivating effects of rewards differ when embedded in a game and when isolated in shell-games. The mathematical game that was used in the current study can be considered a "shell-game", since the rewards were not integrated with each task (i.e. calculation). Future studies can focus on possible differences in the effects of rewards in

both integrated and shell games. We expect that monetary rewards are more effective in shell games compared to embedded games, since they have an external value outside the game.

Alignment of a reward to the transfer effect and user-context of a persuasive game will inform design decisions as to the most optimal reward type, form, and interaction structure for a given player and context. The present paper presented a start in fundamental research on the motivational effect of game-based rewards in persuasive games. Since rewards are so fundamental for human behavior and motivation, and thus for persuasive game research, future research is strongly welcomed, which on the one hand elaborates on reward design (e.g., reward (sub)types, combinations, and interactive structure), and on the other hand on users (e.g., personality and context of use).

**Author Contributions:** Conceptualization, V.T.V., R.S., and M.M.M.v.D.; methodology, R.S.; validation, R.S.; writing—original draft preparation, M.M.M.v.D.; writing—review and editing, M.M.M.v.D., V.T.V., and R.S.; supervision, V.T.V., and R.S.; project administration, V.T.V.; funding acquisition, V.T.V., and R.S.

**Funding:** This research was funded by Dutch governmental funding of the CRISP grant on "G-Motiv project: Designing Motivation: Changing Human Behavior Using Game-Elements" grant, as well as by the NWO and ClickNL Creative Industry grant for the application "NextLevel project: Gamedesign Principles for Effective e-health-based Mental Healthcare Therapy".

**Acknowledgments:** We gratefully express our thanks for support and participation in the reported experiment of the Mistral clinic Brijder Jeugd, ROC Delft, Mildred Valkonet, Ellis Bartholomeus, Ivo Salters, and Ed Tan.

**Conflicts of Interest:** The authors declare no conflict of interest. The funders had no role in the design of the study; in the collection, analyses, or interpretation of data; in the writing of the manuscript, or in the decision to publish the results.

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
