# Peer review of "The Design and Application of Game Rewards in Youth Addiction Care"

_information, doi:10.3390/info10040126_

Round 1
Reviewer 1 Report
This manuscript discusses a study focussing on the use and efficacy of different reward types in persuasive games between substance dependent and non-dependent participants. Using measures of play persistence and reward evaluation, the researchers find that substance dependent participants were not less motivated by reward types than non-substance dependent participants. In fact, substance dependent participants played significantly longer for monetary rewards than non-substance dependent participants. The researchers argue that their results suggest that the use of game-based rewards in persuasive games hold promise for the treatment of substance use disorders.
The manuscript is carefully researched and well written but the information provided about the participants and the statistical analyses need to be improved to bolster the reader's confidence in the results. I provide specific suggestions about how to do this, below.
1) The authors mention that due to computer problems they had to exclude 23 participants from the sample. In other words, the authors lost about 1/3 of their sample due to computer problems, which is highly unfortunate given the already small sample size of the study. It would be useful if the authors elaborate on the computer problems they experienced and they should statistically compare the people who had to be excluded to those who remained in the sample to ensure that there are no meaningful differences between these groups.
2) Building on the previous comment, the authors should present summary statistics for the two groups (substance dependent and non-substance dependent) and conduct statistical tests to ensure that, other than substance use, the groups are balanced on observable characteristics. In the section on "Participants" it seems as though substance dependent individuals may be significantly older and have significantly fewer females than non-substance dependent individuals and these differences may be driving the results that the authors find.
3) Related to comment 2), the statistical analyses which the authors conduct do not adjust for the participants' demographic characteristics so is unclear whether the results are driven by substance use status or other factors. Thus, the authors should estimate a regression model (OLS should be fine here given the nature of the dependent variable, unless there is a spike at 40 minutes of play time, which would make a Tobit model preferable) of play persistence on substance use status and all of the other demographic characteristics that they collected, e.g., gender, age, race, etc. If play persistence for monetary rewards is still significantly higher for substance dependent relative to non-dependent participants then that will allay fears that this effect is not driven by other demographic characteristics that differ across the two groups. With regard to reward evaluation, the authors should estimate an ordered probit model and include all of the demographic characteristics mentioned above.
4) The authors mention that they do not differentiate the group of substance dependent adolescents according to their main type of substance problem. This makes sense because their sample is already small so the cell sizes would be tiny if they differentiated the group in this way. However, it would be useful if the authors report the percentage of subjects whose main type of substance problem is alcohol, cannabis, etc.
5) The authors provide an interesting discussion on how persuasive games can be used in treatment and how they should be tailored to the characteristics of the patient. However, the reader is left wondering whether it would ever be practical to tailor games to specific individuals. If this is not possible then it would be useful to know how the authors would propose designing games for the treatment of substance use disorder, generally.
Author Response
Dear reviewer 1,
Please find our revised manuscript according to the reviewers' comments uploaded on the website. We have clearly highlighted our revisions in yellow, to ensure that changes are easily visible to the editors and reviewers.
We have also lowered the similarity rate. When comparing the original SEGAH conference paper with the submitted journal manuscript, according to https://copyleaks.com/compare, there is a 27% overlap in the main text.
In addition, we have also provided clearer Figure 1 and Figure 2.
Thank you for your considering our manuscript.
Sincerely,
Marierose van Dooren

Reviewer 2 Report
This is an interesting paper which explored three types of reward systems within a game to understand how this may be experienced differently by adolescents within substance dependency therapy compared to healthy controls. Although this seems like an important context to understand the role of persuasive games, the current findings do not appear to present compelling evidence of the “persuasiveness” of the reward features for supporting behaviour change in this context (as this is not measured). Further, the fact that the findings showed that certain rewards motivated play over others (and there was no difference between sub-samples) is not entirely new insight for the literature. As such, I would be cautious about recommending this suitable for publication. I have included some more specific comments based on my review of the manuscript.
#1. It would be helpful to include a little more on the proposed shared neurological dopamine systems which are said to be present for both reward-based gaming behaviour and substance use (p2 of PDF file- lines 52-54). That is, this statement seems rather speculative with the absence of any citations to support it. Further, it would be useful to include a bit more detail here about these neurological mechanisms to give the reader a context to how these rewards systems are being said to be shared between these otherwise different behaviours.
#2. It would be useful to give the mean (and SD) of the two sub-samples in addition to the detail currently included on the overall range. That is, it is likely the non-clinical sample age mean is lower given that this can only extend up to the age of 16 (or 18 perhaps if post-16 education provision is available at the selected school). There may be age-related differences perhaps in regulation systems between the two samples which may be a confound of the study
#3. Having more information about the specific substance use dependence would be helpful, particularly as the authors specifically mention psychoactive substances in particular as being related to a decreased sensitivity to stimuli which are not related to these substances. However, I note that the authors highlight a limitation is that they did not collect this data (in the discussion). In light of this, the authors would be wise to perhaps instead give an overview of what specific substances are the focus of treatment provision for the specific therapeutic setting chosen for the study
#4. Fixed Ratio schedule reinforcement is used whereby rewards are given after a certain number of correct answers. Perhaps a useful way to develop this research further is to explore different reward schedules, given that variable ration schedule may be more likely to promote certain behavioural outcomes which are different from a fixed schedule.
#5. It is not entirely clear within the procedure whether the three “game sessions” were undertaken within one testing period or whether these extended over a longer period of time. This is important based on the fact that play time is a primary DV.
#6. In the results, it mentions that the control variable “order of reward type” but this is not mentioned within the method. Presumably the order of reward types was counterbalanced across the sample (and that this revealed no issues, as evidenced from the non significant difference presented). However, the details of this counterbalancing (if indeed this was undertaken) should be included within the procedure information
#7. In the Design section it includes details that the study duration was 30-60mins. More clarity is needed on this in respect of the following; 1) is this per game session or overall for the entire testing period? or 2) is this a stipulation of the researchers that the max play time should be 60mins and minimum of 30min or 3) is this based on the functionality of the game is being a pre-determined length? or 4) is this based on the analysis of the sample’s own average/range of play time calculated from the play persistence measure? This is important given that one of the key DVs is a behavioural measure of play persistence. That is, if a pre-determined time is set then it seems to me that play persistence is rather constrained by factors beyond the control of the participant.
#8. Are the means (and SDs) presented in the results for example, in respect of playing time measured in minutes? Without a table of means (or detail of this in the overview of the variables), the reader has to interpret this themselves.
#9. P4- lines 170-171 “Our findings suggest that rewards can be as successfully applied in persuasive game design for substance dependent adolescents as for non-dependent adolescents to enhance motivation for tasks”
This is a rather bold statement to make and should perhaps be toned down. That is, the reference to “persuasive game design” implies that there was some success in the intervention (game) persuading behavioural change, which is not the case. At best, the authors can say that the use of monetary rewards are successful in motivating play behaviour in a game for clinical samples, but there is no evidence of any transfer effect in respect of its impact on therapeutic behavioural change (as this wasn’t measured). The authors should be careful in making bold claims so as they are being more representative of the actual findings.
#10. In the implications section, the authors conclude that rewards motivate substance-use dependent adolescents in the same way as non-clinical adolescents and therefore a eHealth in addiction care context is useful. This is true but I don’t think this proposal is anything new in respect of this literature (including that which the authors review in the introduction) and it is not clear here therefore how the current findings actually are providing any new evidence to how persuasive game rewards are particularly useful for clinical care. Also, given that therapeutic behavioural outcomes were not measured, it is also not clear whether there are indeed any persuasive outcomes of the different types of rewards specifically for clinical care. Again, this doesn’t allow any new insight to be brought into this literature.
Minor comments
#11. P2, line 84- this should be “contained fewer females” rather than “contained less females”
Author Response
Dear reviewer 2,
Please find our revised manuscript according to the reviewers' comments uploaded on the website. We have clearly highlighted our revisions in yellow, to ensure that changes are easily visible to the editors and reviewers.
In addition, we have also provided clearer Figure 1 and Figure 2.
Thank you for your considering our manuscript.
Sincerely,
Marierose van Dooren

Reviewer 3 Report
p.p1 {margin: 0.0px 0.0px 0.0px 0.0px; font: 12.0px 'Helvetica Neue'; color: #454545} p.p2 {margin: 0.0px 0.0px 0.0px 0.0px; font: 12.0px 'Helvetica Neue'; color: #454545; min-height: 15.0px} p.p3 {margin: 0.0px 0.0px 0.0px 0.0px; font: 12.0px 'Helvetica Neue'; color: #454545; min-height: 14.0px}
The paper is clearly written and rich of references, however it frequently makes references to the positive effects of playing games, without considering the implications and possible dark side. A context that should not be neglected, considering the target of players considered.
Persuasive games and gamification are two different topics and work following different logics.
By posing the question as to what extent game-based rewards are suitable in a persuasive game design, the two areas are clearly getting overlapped. Moreover, persuasive games not necessarily get to the extent of affecting behaviours; they often struggle already to get to the point of affecting attitudes and perspective on specific topics.
That said, the fact of including two sample, of which one is composed of a population with dependence opens up a reasoning about dependence and gamble, as well as about the gamespace as a different one than reality that can feed forms of escapism.
Rewards are treated as if they were not only central elements of the gaming activity, but indispensable.
Rewards are treated as if they were not only central elements of the gaming activity, but indispensable. However, this is not supported by the same literature cited in the article bibliography, in which well-known and seminal authors identify feedback and outcomes as nodal elements (Salen and Zimmerman, or Shell, just to name a few).
In general it seems that the bibliography is quite updated, but it is lacking of some key references
The authors are not making any mention to seminal researchers who addressed the topic of persuasiveness and also procedurality as Bogost, Sicart and surroundings; while on the topic of gamification, names as Deterding and Zichermann, Linder and Nacke, should be at least present.
Then Rules of Play was written in 2004 by Katie Salen and Eric Zimmerman. Where is it coming from the third author added by the authors? This lightness denotes little knowledge of the field of reference known as game studies/game design, of which Salen and Zimmerman are central figures.
Once again, the reference to monetary rewards should bring to a reflection in terms of gambling, rather than game design or gamification (as it is social recognition).
The game mechanics are quite far from what the field identifies with the rhetoric of persuasive games. Or at least it seems that the authors are using the term with a different meaning than that used in the literature. See “Persuasive Games” by Bogost (2007).
The activity described is rather closer to gamification than persuasive games.
The methodology is clear. What about the persistency of results in time (longitudinal study)?
The article could benefit of a rumination about the need of rewards in general, especially what happens in terms of conveying meanings with bare game mechanics rather than with games based on narrative, as the latter feed engagement and immersion.
Why did you chose such mechanics? On which basis did you design such games?
L180-183 “The finding that participants with substance dependence played longer for monetary rewards was contrary to our expectations. We expected that participants with substance dependence would play shorter for any reward type during the experiment, as research showed that substance dependent individuals have an overall decreased reward sensitivity [36].“ >> the theories about gambling are getting totally along with your results, because players with addiction do need to test and affirm their ability to control.
Abstract: the sentence L 9-10 “substance dependent participants were not less motivated by reward types than non-substance dependent participants were.” should be revised since not it is tangled.
Author Response
Dear reviewer 3,
Please find our revised manuscript according to the reviewers' comments uploaded on the website. We have clearly highlighted our revisions in yellow, to ensure that changes are easily visible to the editors and reviewers.
In addition, we have also provided clearer Figure 1 and Figure 2.
Thank you for your considering our manuscript.
Sincerely,
Marierose van Dooren

Round 2
Reviewer 1 Report
Thank you for attending to my comments on the previous draft, your edits go a long way to improving the manuscript. However, there are still two outstanding issues that I think need to be dealt with prior to publication.
1) It is still not clear to the reader why 1/3 of the sample was dropped. You mention that, " Approximately 50% of the participants that experienced computer problems unwillingly pressed the stop-button while playing ...." What about the other 50% of participants? Was there a problem with the software that you designed, which caused it to crash? Or did some participants decide to withdraw from the study? I am pressing this point because it arouses needless suspicion to refer generically to "computer problems" and then only explain 50% of these problems. When a large proportion of a dataset is dropped, this raises the spectre of sample selection bias so this issue needs to be dealt with directly. This is why I originally suggested you (statistically) compare the group that experienced problems to the group that did not, to see whether they differ according to observable characteristics, e.g., gender, etc. If they don't then you can speak to the generality of your results among samples of this nature. If they do, then all of your results need to be qualified.
2) As per my original suggestion, it is necessary to estimate a statistical model of play persistence and reward evaluation to draw appropriate statistical inferences. As you recognise, you cannot control for gender in nonparametric tests, which is why I suggested you estimate an OLS or tobit model of play persistence and an ordered probit/logit model for reward evaluation. The importance of doing so cannot be understated: any bivariate relationship between, say, play persistence and substance dependence needs to be robust to other factors that may mediate this relationship and that is why investigating this relationship in a multivariate context is necessary. Just because there are no statistically significant differences between male and female participants in their time playing for money rewards does not imply that gender does not influence the relationship between, say, play persistence and substance dependence, particularly given the marked skew in gender across your substance dependent and non-dependent groups.
One additional minor point is included below:
1) You report the age ranges of the substance dependent and non-dependent groups and then take a simple average of the min and max to report the mean. This strikes the reader as inappropriate because presumably there are far more 16-22 year old substance dependent participants compared to 12-16 year old substance dependent patients? If you do not have the actual age of each participant then you should report this as a limitation of the study because, as per the comment above, age may mediate the relationship between, say, play persistence and substance dependence which means you potentially suffer from omitted variable bias in your analyses.
Author Response
Dear reviewer 1,
Please see our response to your comments attached.

Reviewer 2 Report
The authors have addressed previous comments sufficiently.
Author Response
Dear reviewer 2,
We have revised the English of the paper.

Reviewer 3 Report
Thanks for improving the paper answering to the various requests advanced. Now it's much clearer.
Round 3
Reviewer 1 Report
Thank you for attending to my comments on the previous draft, the changes you have made are satisfactory and go a long way to supporting the conclusions you draw.